# Phylotranscriptomic and Evolutionary Analyses of the Green Algal Order Chaetophorales (Chlorophyceae, Chlorophyta)

**DOI:** 10.3390/genes13081389

**Published:** 2022-08-04

**Authors:** Benwen Liu, Yangliang Chen, Huan Zhu, Guoxiang Liu

**Affiliations:** 1Key Laboratory of Algal Biology, Institute of Hydrobiology, Chinese Academy of Sciences, Wuhan 430072, China; 2University of Chinese Academy of Sciences, Beijing 100039, China

**Keywords:** Chaetophorales, divergence time, gene family evolution, phylotranscriptomic analysis, substitution rates

## Abstract

Considering the phylogenetic differences in the taxonomic framework of the Chaetophorales as determined by the use of nuclear molecular markers or chloroplast genes, the current study was the first to use phylotranscriptomic analyses comparing the transcriptomes of 12 Chaetophorales algal species. The results showed that a total of 240,133 gene families and 143 single-copy orthogroups were identified. Based on the single-copy orthogroups, supergene analysis and the coalescent-based approach were adopted to perform phylotranscriptomic analysis of the Chaetophorales. The phylogenetic relationships of most species were consistent with those of phylogenetic analyses based on the chloroplast genome data rather than nuclear molecular markers. The Schizomeriaceae and the Aphanochaetaceae clustered into a well-resolved basal clade in the Chaetophorales by either strategy. Evolutionary analyses of divergence time and substitution rate also revealed that the closest relationships existed between the Schizomeriaceae and Aphanochaetaceae. All species in the Chaetophorales exhibited a large number of expanded and contracted gene families, in particular the common ancestor of the Schizomeriaceae and Aphanochaetaceae. The only terrestrial alga, *Fritschiella tuberosa*, had the greatest number of expanded gene families, which were associated with increased fatty acid biosynthesis. Phylotranscriptomic and evolutionary analyses all robustly identified the unique taxonomic relationship of Chaetophorales consistent with chloroplast genome data, proving the advantages of high-throughput data in phylogeny.

## 1. Introduction

Filamentous green algae of the order Chaetophorales, one of the members of the OCC (Oedogoniales, Chaetophorales, and Chaetopeltidales) clade in the Chlorophyceae (Chlorophyta), have significant research and practical applications [1,2,3,4,5,6]. Recently, a series of studies has made this previously less well-known algal order, the Chaetophorales, increasingly more familiar to phycologists. The Chaetophorales has been shown to exhibit uncontested monophyly [7,8], although some key relationships within this order warrant further investigation. Based on only a few species or a single molecular marker, previous phylogenetic analyses had failed to reveal relationships within the Chaetophorales [9,10,11].

When Caisová et al. [7] published a few additional representative species and *18S rDNA* sequences in the Chaetophorales, the preliminary taxonomic relationship was finally shown. Subsequently, the family Barrancaceae was included as a new member of the Chaetophorales [8], with the broadly defined family, the Chaetophoraceae, being revised with an additional family, the Fritschiellaceae [12].

The six families Schizomeridaceae, Aphanochaetaceae, Barrancaceae, Uronemataceae, Fritschiellaceae, and Chaetophoraceae were included in the order Chaetophorales at the moment, but relationships among this families remain unclear. Based on chloroplast genomes, Liu et al. [13] had successfully determined the taxonomic relationship within the order Chaetophorales, although this was in conflict with the results based on *18S rDNA* genes [13].

In view of the inconsistency of the taxonomic framework of the Chaetophorales based on nuclear molecular markers or chloroplast genome data, we planned to use transcriptomic data to explore the phylogenetic relationships within this order. Due to its distinct benefits, phylotranscriptomics is increasingly being used for phylogenetic and evolutionary investigations on green algae. It has been demonstrated that acquiring high-density transcriptomic data in conjunction with the development of numerous sophisticated analytical techniques is an effective strategy for clarifying ambiguous evolutionary relationships between distinct species [14,15,16,17,18,19,20]. However, no transcriptomic data had been published on members of the Chaetophorales and little was known about the phylotranscriptomics of this order. 

Many studies on the origin, adaptation, and evolution of different species have made use of the expansion and contraction of gene families analysis [18,21,22,23,24,25]. We can better comprehend the evolutionary transitions in this group of algae by analyzing the evolution of gene families. The evolution of organismal diversity has mostly been shaped by natural selection [26]; generally, most genes have been subject to purifying selection in order to maintain their function [27]. However, changes in selection pressure types could lead to evolutionary innovations. The non-synonymous/synonymous (dN/dS) ratio is a measure of the predominant natural selection pressure acting on protein-encoding genes, presented as the ratio of substitution rates at dN and dS sites. Normally, positive selection is shown by values of dN/dS > 1, neutral evolution shown by values of dN/dS = 1 and negative purifying shown by values of dN/dS < 1 respectively [28]. Furthermore, dN/dS ratio analysis could contribute to revealing the evolutionary relationships in this group of algae. The current study intended to advance our current understanding of the evolution of algae in this order, determine the taxonomic relationship, and phylogenetically reconstruct the order Chaetophorales.

## 2. Materials and Methods

### 2.1. Culture Conditions

Twelve cultures representing five families of the Chaetophorales, with the exception of the Barrancaceae (Table 1), were obtained from the Culture Collection of Freshwater Algae at the Institute of Hydrobiology, Chinese Academy of Sciences (FACHB). All algae were cultured in Bold’s Basal Medium (BBM) liquid medium [29] under a photon fluence rate of 15–35 μmol m^–2^ s^–1^ in a 14-h:10-h light: dark cycle at 20 °C.

### 2.2. Library Preparation and Sequencing

Thermo Fisher Scientific’s TRIzol reagent was used to extract total RNA, and oligo-dT magnetic beads were used to separate poly (A) + mRNA. The mRNA was fragmented by divalent cations at a high temperature in NEBNext First Strand Synthe-sis Reaction Buffer (5×), which was as a template to synthesis random hexamer-primed first-strand cDNA and followed by second strand of cDNA. The NEBNext Ultra RNA Library Prep Kit for Illumina (NEB, Lawrence, MA, USA) was used to create the sequencing library, and the NovaSeq 6000 platform was used for the sequencing (Illumina, San Diego, CA, USA).

### 2.3. Quality Control, De Novo Assembly, and Sequence Annotation

Initially, FastQC v0.11.6 (http://www.bioinformatics.babraham.ac.uk/projects/fastqc/, accessed on 12 July 2021) was used to assess the quality of the raw reads, and Trimmomatic v0.38 [30] was used to do quality control on the raw reads. The clean reads were then assembled by using Trinity v2.8.4 [31] and redundant transcripts were removed using CD-HIT v4.7 [32] with default settings.

BUSCO v3.0.2 was used to evaluate the final transcripts’ completeness [33]. Then, each transcript’s open reading frame (ORF) was predicted using TransDecoder v. 5.5.0. Diamond v0.8.22.84 [34] and Pfam searches for the longest ORFs in the Uniref90 database were performed using BLASTP [35] to search for the longest coding region in each transcript using HMMER v3.1b2 [36]. The nucleotide coding sequences (CDS) and amino acid sequences (PEP sequence) of these regions were employed for the future studies after TransDecoder had integrated the BLASTP and Pfam search results into coding sections (Section 2.4 and Section 2.5).

### 2.4. Orthologous Group Identification and Phylotranscriptomic Analysis

Single-copy orthologues were found using OrthoFinder v2.3.12 [37]. For phylotranscriptomic analysis, single-copy orthologue PEP sequences were chosen. Using MAFFT v7.394 [38] and the options -maxiterate 1000 and -globalpair, each single-copy orthologue was aligned. Using the parameter automated1, trimAl v1.4 [39] was used to trim regions with poor alignment. The following phylotranscriptomic analysis employed the orthologous group’s trimmed alignments.

The phylogenetic trees were built using coalescent-based analysis and supergene analysis. SequenceMatrix [40] was used to concatenate all orthologous groups for the supergene analysis, and PartitionFinder 2 [41] was utilized to identify the evolutionary models and the partitioning of the PEP dataset. The PEP dataset was used to estimate phylogenies using maximum likelihood (ML) and Bayesian inference (BI) techniques. Phylogenetic trees were created using MrBayes v3.2.6 and RAxML v8.2.10 [42,43]. With trees sampled every 1000 generations in a Markov chain Monte Carlo (MCMC) study, four Markov chains (three warm, one cold) were used, and 1000 copies of the ML dataset were used in bootstrap studies to determine statistical reliability. For the coalescent-based studies, each single-copy orthologue underwent ML analysis in RAxML using the PROTGAMMAGTR model and 1000 quick bootstrap replicates. With ASTRAL v5.6.3, the coalescence-based species tree (ST) phylogeny was inferred using the best trees [44].

### 2.5. Gene Family Expansion and Contraction Analysis

The CAFE v3.1 algorithm [45] was used to determine the expansion and contraction of gene families based on the estimates, and the significantly modified gene families were identified using the *p*-value threshold of 0.05.

### 2.6. Divergence Time Estimation

Based on the fossil time of the species *F**. tuberosa* vs. *D. mutabilis* and of *D. mutabilis* vs. *Chlamydomonas reinhardtii* in the official website of TIMETREE (http://www.timetree.org/, accessed on 15 July 2022) and the topology of phylogenetic tree, PAML v4.9 [28] was used to estimate divergence time (mcmctree, nsample = 1,000,000; burin = 200,000; seqtype = 0; model = 4).

### 2.7. Substitution Rate Estimation

All single-copy orthologues’ substitution rates and dN/dS ratios were calculated using the CODEML program of PAML v4.9 [28] (runmode = 2, CodonFreq = 2). Orthologues with dS values greater than five were excluded from further analysis.

## 3. Results

### 3.1. De Novo Transcriptome Assembly and Orthology Detection

This study involved transcriptome data from 12 species from eight genera of five families, representing the existing major branches of the Chaetophorales. Information regarding the transcriptomes of Chaetophorales members are summarised in Table 1. According to Illumina paired-end sequencing technology, each species’ raw reads varied from 53,719,904 to 87,268,606 bases. whereas clean reads ranged from 53,250,356 bp to 87,168,808 bp after filtering out adapters and low-quality sequences. De novo assembly revealed that the N50 values of all species ranged from 1461 bp to 2530 bp, indicating high contiguity. According to searches for the BUSCOs (Benchmarking Universal Single-Copy Orthologues) established for the Chlorophyta, the percentages of conserved genes in our 12 transcriptomes were >80%, reflecting a high level of completeness. TransDecoder predicted between 19,298 and 68,361 bases of coding sequences.

The CDS and PEP of the coding sequences of the 12 species, along with the corresponding *C. reinhardtii* sequences retrieved from the NCBI database, were utilized to search for orthologs. There were 487,252 genes in total and were divided into orthogroups. There were 17,553 orthogroups, and all of them were employed for the examination of the evolution of gene families. Each orthogroup represented a family of genes. For the phylotranscriptomic study, estimation of substitution rate, and estimation of divergence time, a total of 143 single-copy orthogroups were used.

### 3.2. Phylotranscriptomic Analysis

Based on the PEP sequences of 143 single-copy orthogroups, the ML and BI phylogenetic trees (Figure 1) for all 13 species were created. In contrast to earlier analyses of rDNA datasets [7,8,46], supergene analysis and coalescent-based analysis produced the same topology of the Chaetophorales with robust support at almost all nodes, but they were compatible with the phylogenetic tree based on the chloroplast protein-coding genes [13]. With the exception of the Barrancaceae, the five currently recognized families—Schimeriaceae, Aphanochaetaceae, Uronemataceae, Fritschiellaceae, and Chaetophoraceae—diverged into four well-supported clades within the Chaetophorales. At the base of the order Chaetophorales, the Schizomeriaceae and Aphanochaetaceae were grouped together. The Uronemataceae, as the sister, had the closest relationship with the clade comprising the Fritschiellaceae and Chaetophoraceae, which was located at the top branch of the Chaetophorales. The genus *Stigeoclonium* was still polyphyletic.

### 3.3. Gene Family Expansion and Contraction Analysis

The CAFE computational tool’s birth-death model, which makes the assumption that at least one gene resides at the species tree’s root, was used to analyze the evolution of gene families. Figure 2 depicts the numerous expansions and contractions that the gene families of all Chaetophorales had undergone. A total of 525 gene families exhibited expansion compared with 3456 gene families undergoing contraction. *S**. leibleinii* displayed 3456 contracted gene families and *F. tuberosa* (the only terrestrial alga in the Chaetophorales) exhibited 2284 expanded gene families, numbers which were greater than those in all other nodes and species.

Using gene ontology (GO) enrichment and the Kyoto Encyclopedia of Genes and Genomes (KEGG) pathway analyses, the roles of the enlarged gene families of *F. tuberosa* were examined. One of the gene families that underwent significant expansion in *F. tuberosa* (adjusted *p* < 0.01) was primarily involved in fatty acid biosynthesis (Ko00061) (Figure 3a), a finding which was consistent with the high production of fatty acids by this terrestrial alga (Figure 3b,c).

### 3.4. Divergence Time Estimation

The tree was constructed based on 143 single-copy orthologues. The average divergence times of most species were in the range 586–893 MYA, located in the Proterozoic era, during which period the biological world evolved from prokaryotes to eukaryotes, being dominated by fungus and algae, with the Schizomeriaceae, Aphanochaetaceae, and Fritschiellaceae diverging later and all of them located in the Paleozoic era. Divergence times between the clade (Schizomeriaceae + Aphanochaetaceae) and the other families of the Chaetophorales were about 743–962 MYA, whereas divergence time between the Schizomeriaceae and the Aphanochaetaceae was about 357 MYA. Divergence time between *Stigeoclonium* sp. (SRR19138393) and *F. tuberosa* (SRR19137783) was about 367 MYA. *A. confervicola* (SRR19135353) and *A. elegans* (SRR19135358) did not diverge until the Mesozoic era (Figure 4).

### 3.5. Substitution Rate Estimation

The rate of dN and dS substitutions was calculated utilizing the ML method. There were 140 single-copy orthogroups after the orthogroups with dS > 5 were eliminated. The dN and dS rates of each species are shown in Figure 5a,b.

The dN and dS rates of *A. confervicola* were higher than those of other species. We selected species representative of each family for significance testing. *S. leibleinii* (family Schizomeriaceae) and *A. confervicola* (family Aphanochaetaceae) exhibited no significant difference with respect to either dN or dS rates. *S. leibleinii* (family Schizomeriaceae) was significantly different from *U. confervicolum* (family Uronemataceae) and *D. mutabilis* (family Chaetophoraceae). Families of the Uronemataceae, Fritschiellaceae, and Chaetophoraceae were not significantly different from one another.

## 4. Discussion

Previous studies based on nuclear markers had clearly shown the Schizomeridaceae, to be the basal clade, being a sister of all of the other remaining families of the Chaetophorales and being markedly separate from the Aphanochaetaceae with robust support [7,8,12,13]. On the other hand, phylogenetic analyses based on the chloroplast protein-coding genes showed a different scenario, in that the Schizomeriaceae and Aphanochaetaceae could not be clearly separated in terms of rDNA datasets and were instead, clustered into one branch at the base of the order Chaetophorales [13]. In the present study, phylotranscriptomic analyses based on 143 single-copy orthologues strongly supported the latter results, with the Schizomeriaceae and Aphanochaetaceae clustered into one branch at the base of the Chaetophorales order, findings which were consistent with those from phylogenetic analyses based on chloroplast genome data [13].

There may be paralogues, different evolutionary rates, insufficient lineage sorting, horizontal gene transfer, or gene duplication as the primary causes of gene tree discrepancies [47,48,49,50,51,52,53]. Increases in dataset information and the use of an analysis method that is more appropriate could lessen the impact of such errors [54,55,56,57] and enhance the phylogenetic analysis’s support, resolution, and accuracy [58]. Considering the phylogenetic discrepancy between the nuclear molecular markers and the chloroplast genes on the taxonomic framework of the Chaetophorales and the unique advantages of the transcriptome, the current study was the first to subject the transcriptomes of Chaetophorales algae to phylotranscriptomic analyses. In this study, based on supergene concatenated by 143 single-copy orthologues and partitioned to apply different evolutionary models to different groupings, plus coalescent-based analyses approaches, phylotranscriptomic analyses of Chaetophorales proved to be more reliable. The incongruence between gene trees and species trees can be reduced by using a high number of unlinked loci in phylotranscriptomic analyses based on single-copy orthologues [18,59,60,61]. Although high levels of gene tree conflicts for incomplete lineage sorting [57] or horizontal gene transfer [55,56], based on coalescent-based method, ASTRAL was used to estimated species trees from multiple genes resulted in highly accurate phylogenomic estimation.

Different variables, such as gene duplication, de novo gene production, gene loss, the function of gene families, and changes in environmental conditions, may have an impact on the number of expanded or contracted gene families [62,63,64]. The finding of expanded gene families of *F. tuberosa* associated with the functions of fatty acid biosynthesis was common among numerous terrestrial algae and reflected a tolerance strategy against stressful environments [65]. One of the causes of genome decrease, according to a prior study, was gene family contraction [66]. Unfortunately, no whole genome of a member of the Chaetophorales has been published to test this hypothesis.

Phylogenetic analysis and divergence time estimation was performed among twelve species of green algae of the order Chaetophorales. In the Proterozoic era, many fossils of fungi, algae, and ancient microbes have been found. Thus, this era was also called the era of bacteria and algae. The average divergence times of most species were located in this special period, which was consistent with the findings for green algae of this period. The clade comprising the Schizomeriaceae and Aphanochaetaceae diverged earlier and the two families subsequently diverged from one another, indicating the close relationships between them. Fossil evidence suggested that, around 500–600 MYA, a type of green alga successfully resisted the arid, terrestrial conditions and took to the land, becoming the ancestor of land plants. As the only terrestrial member of the Chaetophorales, that *F. tuberosa* (SRR19137783) diverged from the aquatic species at 367 million years ago appeared reasonable, since it was not on the evolutionary line from which land plants originated.

Most orthogroups (*n* = 110) had dN/dS ratios greater than 1, indicating positive selection, with the non-synonymous mutations resulting from Darwinian positive selection [67]. Positive selection is closely related to the adaptive evolution of species [17,44,68]. Only 33 of the orthogroups had ratios that were significantly low (dN/dS < 1), indicating strong negative (purifying) selection [28]. The dN and dS rates between the Schizomeriaceae and Aphanochaetaceae were not significantly different from each other, although the clade comprising the Schizomeriaceae and Aphanochaetaceae had dN and dS rates significantly higher than those of other families, which reflected the close relationship between the Schizomeriaceae and the Aphanochaetaceae in the Chaetophorales.

This study provided the first transcriptomic data and determined the taxonomic framework and evolutionary relationships of the green algae Chaetophorsles, and revealed that the expansion gene families of genus *Fritschiella* were related to fatty acid synthesis, which provided a research basis for the utilization of this algae resources. The terrestrial alga *F. tuberosa* is a promising microalga for sustainable co-production of pigments and fatty acids in future.

## 5. Conclusions

Phylotranscriptomic analysis showed that the unique taxonomic scheme of the Chaetophorales obtained was consistent with that obtained from chloroplast genome data, a finding which proved the advantages of high-throughput data in phylogeny and gave new insights to the evolution of the Chaetophorales. Unfortunately, little transcriptomic and genomic data of the Chaetophorales have been published to date and further study is needed.

## Figures and Tables

**Figure 1 genes-13-01389-f001:**
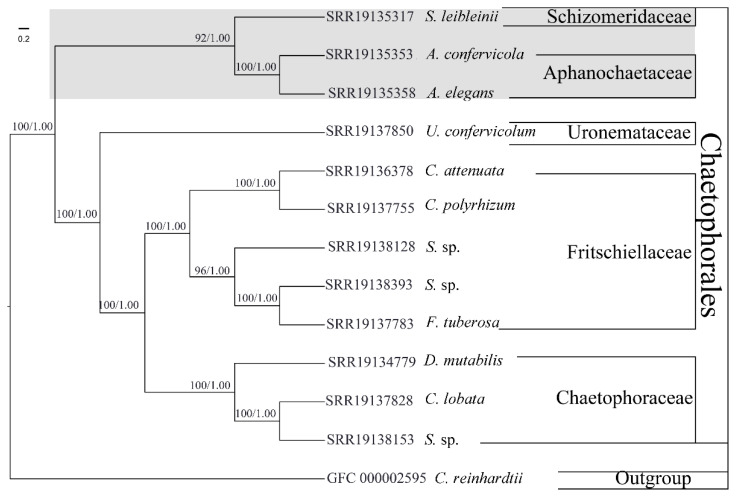
143 single-copy orthogroups were used to construct the phylogenetic tree for the order Chaetophorales. The numbers on the nodes correspond to the bootstrap support values (BP)/posterior probabilities (PP). The scale bar represents the genetic distances, which are proportional to branch lengths. The Schizomeridaceae and Aphanochaetaceae family clade is indicated by the gray backdrop.

**Figure 2 genes-13-01389-f002:**
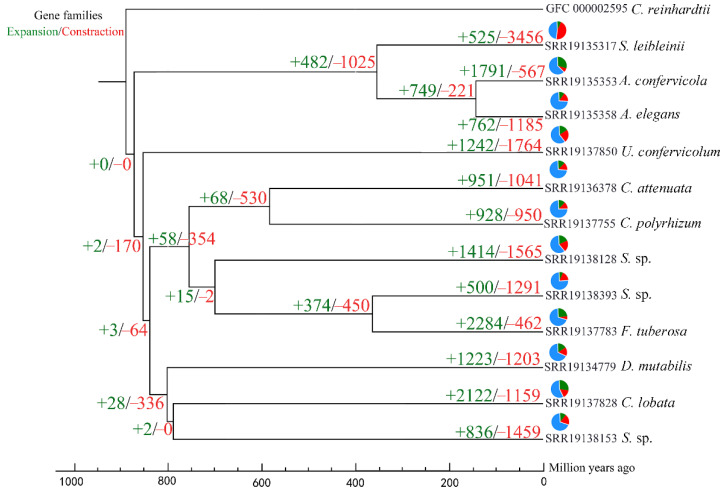
The branch numbers represent the number of gene families that have expanded (+) and contracted (−) since the common ancestor’s split. The relative ratio of expansion or contraction is shown on the pie chart. The timelines show the times at which the species diverged.

**Figure 3 genes-13-01389-f003:**
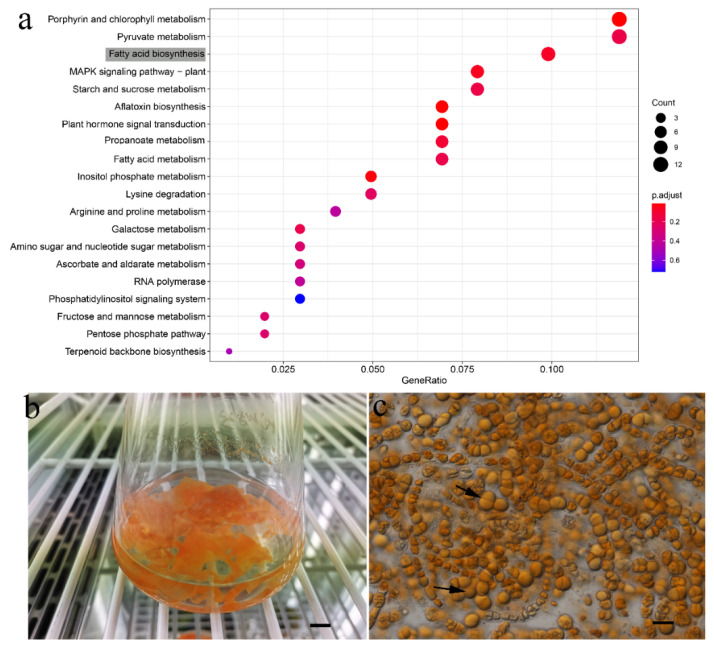
(**a**). Dot plot displaying the enrichment of *F. tuberosa*’s expanded gene families. The numbers of genes were represented by dot sizes. Fatty acid biosynthesis was in grey. (**b**) Shows the color change of the *F. tuberosa* in culture. (**c**) The arrow indicates the fatty acid. Scale bars: b = 1 cm; c = 20 μm.

**Figure 4 genes-13-01389-f004:**
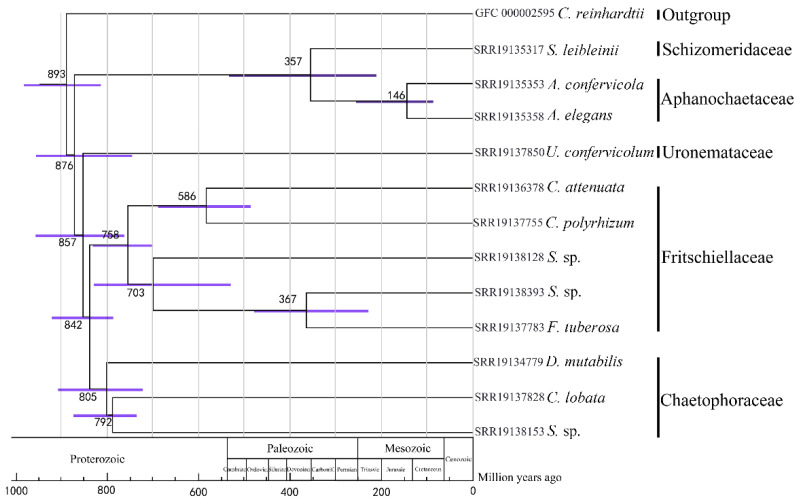
Divergence time estimated by fossil calibration. Node numbers represented estimated time (Mya). The purple bar represented range of 95% HPD.

**Figure 5 genes-13-01389-f005:**
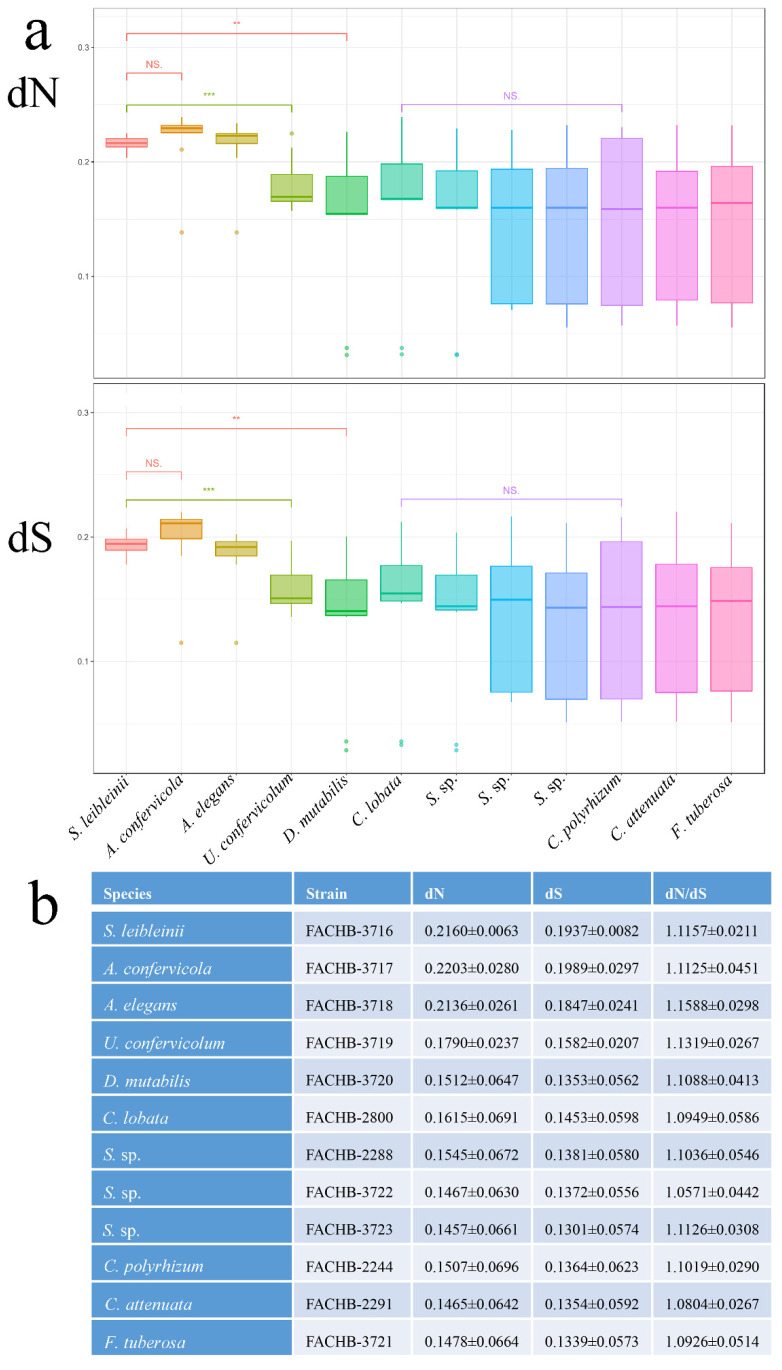
(**a**). Boxplots showing non-synonymous (dN) and synonymous (dS) substitutions of the Chaetophorales. ** 0.001 < *p* < 0.01, *** *p* < 0.001, NS., non-significant. (**b**) The average value and standard deviation of dN, dS and dN/dS for each species.

**Table 1 genes-13-01389-t001:** Summary of transcriptome data of the Chaetophorales.

Species	Voucher	Culture	No. of Raw Reads	No. of Clean Reads	N50 Length	Single-Copy Orthologs	Complete BUSCOs (%)	Number of Coding Sequence
*Schizomeris leibleinii*	HB202102	FACHB-3716	53,719,904	53,250,356	2530	1531	86.40	19298
*Aphanochaete confervicola*	HB201725	FACHB-3717	54,935,394	54,925,846	1461	1023	87.00	68361
*Aphanochaete elegans*	HB201732	FACHB-3718	54,067,336	54,055,980	1864	1360	84.10	42264
*Uronema confervicolum*	LY201701	FACHB-3719	58,939,480	58,861,738	1937	1495	85.40	41588
*Draparnaldia mulabilis*	AES201713	FACHB-3720	76,239,588	76,226,452	1614	1472	83.40	31263
*Chaetophora lobata*	QH201901	FACHB-2800	55,627,796	55,487,446	1544	1186	81.60	49304
*Stigeoclonium* sp.	HB201635	FACHB-2288	68,612,258	68,599,684	2055	1690	82.30	29126
*F. tuberosa*	HB201823	FACHB-3721	58,548,230	58,465,136	1657	1308	82.80	55524
*Chaetophoropsis polyrhizum*	HB201646	FACHB-2244	60,022,402	60,008,734	2090	1634	83.20	34764
*Stigeoclonium* sp.	HB201648	FACHB-3722	60,955,222	60,946,270	1946	1471	83.50	31683
*Stigeoclonium* sp.	YN201601	FACHB-3723	63,709,968	63,690,832	1990	1762	83.70	32184
*Chaetophoropsis attenuata*	FHB201644	FACHB-2291	87,268,606	87,168,808	1720	1621	81.80	34151

## Data Availability

The raw data was submitted to the National Center for Biotechnology Information (NCBI) at https://www.ncbi.nlm.nih.gov/ (accessed on 18 May 2022) and Sequence Reads Archive database under BioProject accession number PRJNA835257.

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
