# Peer review of "Phylotranscriptomic and Evolutionary Analyses of the Green Algal Order Chaetophorales (Chlorophyceae, Chlorophyta)"

_genes, 2022, doi:10.3390/genes13081389_

Round 1

Reviewer 1 Report

The paper is really interesting and it paves the way for future transcriptomic analysis to take full advantage of the algae strains.

The paper is very well prepared.

My only regret is about the missing information of what kind of fatty acids are produced by the studied strain, what are their potential applications.

Also, the color of the strain is very pronounced. Do you know which kind of compounds are responsible for this orange color (carotenoids?) and what could be their applications? Did you identified some GO associated with the production of these colored compounds?

To summarize a few more discussion about the potential applications of the present work and the studied strain would be welcome and further improve the already high quality of this work.

Author Response

Dear editor and anonymous reviewers,

Thank you for your letter and comments on our manuscript (Manuscript ID: genes-1841302) dated 14 July 2022. Based on your comments and requests, we have made modification and revised our manuscript. Here, we submitted our revised manuscript for your approval. Our answers of every questions point by point from you and the reviewers are as follows:

Reviewer #1:

  1. The paper is really interesting and it paves the way for future transcriptomic analysis to take full advantage of the algae strains. The paper is very well prepared.

Re: Thank you for your affirmation of our MS.

  1. My only regret is about the missing information of what kind of fatty acids are produced by the studied strain, what are their potential applications.

Re: Yes, indeed so. The types of fatty acids are really interesting. Originally, our focus is on phylogenetic and evolutionary analyses of the order Chaetophorales. I am sorry.

Our team previously analyzed the oil production of filamentous green microalga Oedocladium carolinianum, the terrestrial alga as the Fritschiella tuberosa. The potential for a high lipid content in Fritschiella tuberosa may offer a great opportunity for this organism to be used as an oil-rich feedstock and for animal feed application.

  1. Also, the color of the strain is very pronounced. Do you know which kind of compounds are responsible for this orange color (carotenoids?) and what could be their applications? Did you identified some GO associated with the production of these colored compounds?

Re: Thank you for your comments. I'm not sure that is the real compounds responsible for this orange color. I am sorry. It may be astaxanthin, canthaxanthin and lutein or other pigments. Some terrestrial filamentous green microalgae like Fritschiella tuberosa, the Oedocladium carolinianum analyzed by our team is astaxanthin-producing (See reference Wang et al., 2022) while the Barranca yajiagengensis (in the same order Chaetophorales with Fritschiella tuberosa) is canthaxanthin-producing and lutein-producing (See reference Gao et al., 2022). We will conduct this study in future.

Few GO associated with the production of these colored compounds has been identified. It has been reported that greater amounts of astaxanthin or canthaxanthin are produced under conditions of stress when the nutrient availability is no longer adequate such as under high light intensity or nitrogen starvation. In present study, however, all strains used for phylogenetic and evolutionary analyses were harvested at good culture conditions which could be the main reason why GO associated with the production of these colored compounds have not been identified.

  1. To summarize a few more discussion about the potential applications of the present work and the studied strain would be welcome and further improve the already high quality of this work.

Re: Thank you for your suggestions. We have added some more discussion about potential applications of the present work and the studied strain in this MS. Please see response below and file genes-1841302_R1_Liu_edit_manuscript track changes.

This study provided the first transcriptomic data and determined the taxonomic framework and evolutionary relationships of the green algae Chaetophorsles, and revealed that the expansion gene families of genus Fritschiella were related to fatty acid synthesis, which provided a research basis for the utilization of this algae resources. F. tuberosa is a promising microalga for sustainable co-production of pigments and fatty acids in future.

Reviewer 2 Report

Liu et al. reported a timely and relevant investigation in the manuscript entitled “Phylotranscriptomic and evolutionary analyses of the green agal order Chaetophorales (Chlorophyceae, Chlorophyta)”. I would request the authors to address the following concerns:

1.     Title: agal or algal?

2.     I could not access data at PRJNA835257. Is the submission private at this moment? If yes, please make it public or double-check the submission.

3.     What is the rationale behind selecting only one representative from Schizomeriaceae and two from Aphanochaetaceae? If these families could not be adequately separated, then why not take more? Please add this rationale to the manuscript.

4.     Figure 3b, c: The figure needs to be explained in the text of the manuscript and how is it related to the significance of the study.

5.     Is it possible to calculate and tabulate dN/dS ratios for each of the species represented in figure 5?

Author Response

Dear editor and anonymous reviewers,

Thank you for your letter and comments on our manuscript (Manuscript ID: genes-1841302) dated 14 July 2022. Based on your comments and requests, we have made modification and revised our manuscript. Here, we submitted our revised manuscript for your approval. Our answers of every questions point by point from you and the reviewers are as follows:

Reviewer #2:

  1. Title: agal or algal?

Re: Thank you for your comments. I am sorry for such mistake. I have fixed it.

  1. I could not access data at PRJNA835257. Is the submission private at this moment? If yes, please make it public or double-check the submission.

Re: Thank you for your suggestions. The released date has been set when I submitted this data to NCBI. I am sorry. This BioProject submission will be released on 2022-12-31 or upon publication, whichever is first. I has emailed the staffs of NCBI.

  1. What is the rationale behind selecting only one representative from Schizomeriaceae and two from Aphanochaetaceae? If these families could not be adequately separated, then why not take more? Please add this rationale to the manuscript.

Re: Thank you for your comments. Until now, the family Schizomeriaceae includes only one genus and one species (the type species S. leibleinii). The family Aphanochaetaceae includes one genus and four species formed two clades in the phylogenetic tree and two representative species (the type species) from each clade were selected for phylotranscriptomic and evolutionary analyses. In my opinion, the reasons why these families could not be adequately separated are not for lack samples but for the characteristics of genes themselves

  1. Figure 3b, c: The figure needs to be explained in the text of the manuscript and how is it related to the significance of the study.

Re: Thank you for your comments. Figure 3b, c showed the fatty acids in the cells to account for the significant expansion fatty acid biosynthesis gene families in the F. tuberosa. Please see file genes-1841302_R1_Liu_edit_manuscript track changes.

  1. Is it possible to calculate and tabulate dN/dS ratios for each of the species represented in figure 5?

Re: Thank you for your suggestions. We have calculated and tabulated dN/dS ratios for each species. Please see figure 5b.

Round 2

Reviewer 2 Report

The authors have revised the manuscript satisfactorily.